# Nonsingular Terminal Sliding Mode Based Finite-Time Dynamic Surface Control for a Quadrotor UAV

Yuxiao Niu [1] , Hanyu Ban [2], Haichao Zhang [1], Wenquan Gong [1,*] and Fang Yu [1]

1   Institute of Logistics Science and Engineering, Shanghai Maritime University, Shanghai 201306, China; niuyuxiao@stu.shmtu.edu.cn (Y.N.); zhanghaizhao@stu.shmtu.edu.cn (H.Z.); yufang@shmtu.edu.cn (F.Y.)
2   Innovation Academy for Microsatellites of CAS, Shanghai 201204, China; banhy@microsate.com
*   Correspondence: dalekwho110@gmail.com

**Abstract:** In this work, a tracking control strategy is developed to achieve finite-time stability of quadrotor Unmanned Aerial Vehicles (UAVs) subject to external disturbances and parameter uncertainties. Firstly, a finite-time extended state observer (ESO) is proposed based on the nonsingular terminal sliding mode variable to estimate external disturbances to the position subsystem. Then, utilizing the information provided by the ESO and the nonsingular terminal sliding mode control (NTSMC) technique, a dynamic surface controller is proposed to achieve finite-time stability of the position subsystem. By conducting a similar step for the attitude subsystem, a finite-time ESO-based dynamic surface controller is proposed to carry out attitude tracking control of the quadrotor UAV. Finally, the performance of the control algorithm is demonstrated via a numerical simulation.

**Keywords:** nonsingular terminal sliding mode control; finite-time stability; quadrotor UAV; dynamic surface control

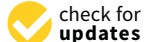



## 1. Introduction

The advantages of Quadrotor Unmanned Aerial Vehicles (UAVs) hare their small size, low energy consumption, and great flexibility. Therefore, they have extensive applications, including environmental monitoring, aerial photography, logistics distribution, and so on. Recently, the quadrotor UAV has been studied extensively. Many scholars have researched control problems regarding the quadrotor UAV, including trajectory tracking control [1], formation control [2], obstacle avoidance control [2], and fault tolerant control [3]. High-accuracy trajectory tracking control is the basis for allowing the quadrotor UAV to complete tasks. Hence, the tracking control trajectory is one of the most important aspects of a quadrotor UAV. However, external disturbances and parameter uncertainties will have negative impacts on the tracking control system. Therefore, studies on the trajectory tracking control problem are becoming increasingly significant for quadrotor UAVs subject to synthetic disturbances [1,4,5].

There are many methods to solve the control problem, including sliding mode control (SMC) [6–8], proportional-integral-derivative (PID) or proportional-derivative (PD) [9] control, adaptive control (AC) [10,11], backstepping control [12,13], dynamic surface control (DSC) [14], neural network [15], and model predictive control [16]. Backstepping control is one of the most effective control methods. In [17], a backstepping-technique-based controller was designed for a typical multi-input multi-output (MIMO) system class to address the tracking problem. In [18], an integral backstepping control strategy was proposed for a quadrotor with unknown modeling uncertainties and disturbances to ensure that the system was asymptotically stable. By utilizing the backstepping and fuzzy control techniques, a new sliding mode controller was presented by [19] and used to determine the robustness of the differential wheeled mobile robot. However, backstepping has the disadvantage of leading to an "explosion of complexity" after multiple iterations. The DSC technique can overcome this drawback by introducing a filter. The filter can estimate

the derivative of the virtual control law. A DSC-based trajectory tracking controller was designed by [20] for a quadrotor UAV by introducing a first-order low-pass filter. However, the aforementioned scheme did not take into account the estimation error of the filter. In [21], an error compensation signal was designed to compensate for the estimation error. To design a backstepping recursive control scheme for MIMO nonlinear systems, a new finite-time filter was proposed to obtain command signals and their derivatives by [22], where the system was practically finite-time stable.

SMC is also one of the most commonly used methods to deal with the tracking control problem. SMC has a history of more than 60 years, and it has the characteristics of simplicity and robustness [23]. SMC has a wide range of applications [24]. However, the bulk of SMC techniques can only ensure asymptotic stability of the system. In order to obtain a fast convergence rate, terminal sliding mode control (TSMC) was developed. In [25], a nonsingular terminal sliding mode control (NTSMC)-based control input was designed to guarantee the fixed-time stability of a second-order nonlinear system. In [26], a continuous integral terminal sliding mode variable was designed to solve the singularity and chattering problems, and the sliding mode variable based control algorithm was proposed to guarantee fixed-time stability of second-order nonlinear systems. To obtain a fast response from the system, a fast terminal sliding mode control (FTSMC)-based controller was designed by [27] for quadrotor UAVs subject to external disturbances and parameter uncertainties. In [28], a FTSMC-based trajectory tracking strategy for autonomous underwater vehicles was proposed to improve the convergence rate. An adaptive PID-SMC method was proposed for quadrotor UAVs subject to external disturbances to achieve finite-time stability of the tracking control system in [29].

In applications, disturbances in the system will reduce the accuracy of the control scheme. One of the most effective ways to overcome this is to use an observer-based approach. To estimate the external disturbances, an AC-based terminal sliding mode observer for quadrotor UAVs was proposed to achieve appointed-fixed-time stability of the attitude system [30]. A disturbance observer (DO) based tracking control strategy was proposed to ensure the asymptotic stability of a wheeled mobile robot in [31]. For typical nonlinear systems, the authors of [32] presented a DO-based control framework to achieve asymptotic stability of the closed-loop tracking system. In [33], a DO was designed for mechanical systems to estimate the total uncertainty, and a controller was proposed to ensure the stability of the system. In [34], an extended state observer (ESO) for robot manipulators was proposed to estimate the velocity measurement uncertainty in finite time. In [35,36], a finite-time ESO was designed based on the NTSMC technique to address the fault tolerant control problem, and this could be used to estimate the lumped uncertainties of the spacecraft. For a quadrotor with disturbances and model uncertainties, a filtered observer-based Interconnection and Damping Assignment-Passivity Based Control scheme was designed to address the tracking problem [37]. Utilizing the filter to attenuate noise, a tracking controller was proposed to achieve asymptotic stability of the closed-loop system for quadrotor UAVs [38].

This paper addresses the tracking control problem associated with quadrotor UAVs subject to external disturbances and parameter uncertainties. The main contributions of this paper are as follows: (1) For the position and attitude subsystems, two NTSMC-based finite-time ESOs are proposed to estimate external disturbances and/or parameter uncertainties. Compared with the asymptotically stable nonlinear ESO presented in [39], the proposed ESO is finite-time stable. The finite-time scheme is not only robust and highly precise, it can also estimate the upper bound of the settling time. (2) Inspired by [14], two finite-time controllers based on NTSMC and DSC are designed for the position and attitude subsystems. The estimation error of the filter converged to a neighborhood of the origin in [14]. The advantage of this paper is that the filter can precisely estimate the derivative of the virtual control law, and the estimation error can converge to zero in finite time. Hence, the convergence rate and the tracking error are improved in the proposed control scheme.

The rest of this paper is organized as follows. In Section 2, the quadrotor model is established. In Section 3, the NTSMC technique-based finite-time ESOs and dynamic surface controllers are proposed, and the stabilities are analyzed in detail using the Lyapunov criteria. A numerical simulation of the proposed control algorithm is performed in Section 4. The conclusions are drawn in Section 5.

**Notation.** The superscript T represents the transpose of a matrix. Denote $\text{sig}^b(x) = |x|^b \text{sign}(x)$, where $\text{sign}(\cdot)$ is a standard symbolic function and $|\cdot|$ is the absolute value. $\|\cdot\|$ stands for the Euclidean norm of a vector. $\mathbb{R}$ and $\mathbb{R}^+$ respectively represent the sets of real numbers and positive real numbers. $\mathbb{R}^n$ denotes an n-dimensional real vector. $\mathbb{U}\backslash\{0\}$ represents the set, where $\{0\}$ is removed from the set $\mathbb{U}$.

## 2. Quadrotor Model

As shown in Figure 1, four motors are used to generate thrust for the quadrotor UAV. The four types of thrust can be expressed as $F_i$ ($i$ = 1, 2, 3, 4), and these can be controlled by adjusting the steering of the motors. The dynamics of the quadrotor UAV are represented as [40]:

$$\begin{cases} \ddot{x} = \frac{u_F}{m}(\cos\phi\sin\theta\cos\psi + \sin\phi\sin\psi) - \frac{k_1}{m}\dot{x} + d_1 \\ \ddot{y} = \frac{u_F}{m}(\cos\phi\sin\theta\sin\psi - \sin\phi\cos\psi) - \frac{k_2}{m}\dot{y} + d_2 \\ \ddot{z} = \frac{u_F}{m}(\cos\phi\cos\theta) - g - \frac{k_3}{m}\dot{z} + d_3 \\ J_1\ddot{\phi} = -k_4 l\dot{\phi} + d_4 + lu_1 - \Delta J_1\ddot{\phi} \\ J_2\ddot{\theta} = -k_5 l\dot{\theta} + d_5 + lu_2 - \Delta J_2\ddot{\theta} \\ J_3\ddot{\psi} = -k_6\dot{\psi} + d_6 + cu_3 - \Delta J_3\ddot{\psi} \end{cases} \quad (1)$$

where $(x, y, z)$ and $(\phi, \theta, \psi)$ represent the position and attitude of the quadrotor UAV, respectively; $\psi \in (-\pi, \pi)$ is yaw; $\theta \in (-\pi/2, \pi/2)$ is pitch; $\phi \in (-\pi/2, \pi/2)$ is roll; $m \in \mathbb{R}^+$ is the mass of the quadrotor UAV; $d_i \in \mathbb{R}(i$ = 1, 2, ..., 6) is the external disturbance; $J_i \in \mathbb{R}^+$ ($i$ = 1, 2, 3) is the moment of inertia; $k_i \in \mathbb{R}^+(i$ = 1, 2, ..., 6) represents the aerodynamic damping coefficient; $\Delta J_i \in \mathbb{R}$ ($i$ = 1, 2, 3) is the uncertainty of the moment of inertia; $l \in \mathbb{R}^+$ is the distance from the motor to the center of mass; $c \in \mathbb{R}^+$ denotes the force-to-moment factor; and $u_F$, $u_1$, $u_2$, $u_3$ are the control inputs of the position and attitude subsystems, respectively.

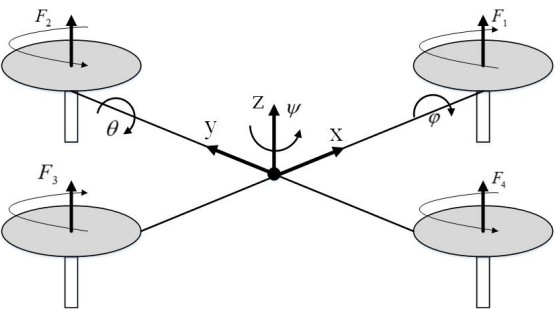

**Figure 1.** The structure of the quadrotor UAV.

**Lemma 1** ([41])**.** *Consider the following system*

$$\dot{\delta} = f(\delta),\ \delta(0) = \delta_0,\ f(0) = 0,\ \delta \in \mathbb{R}^n \quad (2)$$

*where $\delta = [\delta_1, \delta_2, \ldots, \delta_n]^T$ is the state vector. Suppose there is a positive definite Lyapunov function $V(\delta)$, which is defined on a neighborhood $\mathbb{U} \subset \mathbb{R}^n$ of the origin. For any $\delta \in \mathbb{U}\backslash\{0\}$, if the inequality*

$$\dot{V}(\delta) + \nu_1 V(\delta)^p \leqslant 0 \quad (3)$$

is satisfied with $\nu_1 > 0$, $0 < p < 1$. The system is finite-time stable, and the settling time is

$$t_1 \leqslant \frac{1}{\nu_1(1-p)}|V(\delta_0)|^{1-p} \tag{4}$$

**Lemma 2** ([42]). *Consider the system in* (2). *For* $p \in (0,1)$, $\nu_1 > 0$ *and* $\nu_2 > 0$, *suppose a positive definite Lyapunov function* $V(\delta)$ *exists, which satisfies*

$$\dot{V}(\delta) \leqslant -\nu_1 V(\delta) - \nu_2 V(\delta)^p \tag{5}$$

*such that the origin of the system is finite-time stable. The settling time is*

$$t_2 \leqslant \frac{1}{\nu_1(1-p)} \ln \frac{\nu_1 V(\delta_0)^{1-p} + \nu_2}{\nu_2} \tag{6}$$

**Lemma 3** ([43]). *Consider the system in* (2). *If a Lyapunov function* $V(\delta)$ *exists, which satisfies*

$$\dot{V}(\delta) \leqslant -\nu_1 V(\delta)^p + \nu_2 \tag{7}$$

*with* $p \in (0,1)$, $\nu_1 > 0$, $\nu_2 > 0$. *The system is finite-time stable in a neighborhood of the origin. The settling time* $t_3$ *is given by*

$$t_3 \leqslant \frac{V(\delta_0)^{1-p}}{\nu_1 \theta_1(1-p)}, \quad \theta_1 \in (0,1) \tag{8}$$

**Lemma 4** ([25]). *Consider a general second-order system*

$$\begin{cases} \dot{\delta}_1 = \delta_2 \\ \dot{\delta}_2 = f(t,\delta) + g(t,\delta)(u + d(t,\delta)) \end{cases} \tag{9}$$

*where* $\delta = [\delta_1, \delta_2]^{\mathrm{T}}$ *is the state vector;* $u$ *denotes the control input;* $f(t,\delta)$ *and* $g(t,\delta)$ *satisfy* $f(t,0) = 0$ *and* $g(t,\delta) \neq 0$, *respectively; and* $d(t,\delta)$ *represents the lumped disturbance.*

*The following sliding mode variable is designed as*

$$s = \delta_2 + 2\zeta\sqrt{|\arctan(\delta_1)|}(1 + \delta_1^2)\mathrm{sign}(\delta_1) \tag{10}$$

*with* $\zeta > 0$. *If* $s = 0$ *is satisfied, the system will converge to zero in fixed time.*

**Lemma 5.** *Consider the sliding mode variable* (10), *the following nonsingular terminal sliding mode variable tackles the singularity problem.*

$$s = \dot{\delta} + 2\zeta\sqrt{\rho + |\arctan(\delta)|}(1 + \delta^2)\mathrm{sign}(\delta) \tag{11}$$

*where* $\rho$ *is a small constant. The nonsingular terminal sliding mode variable* (11) *is finite-time stable.*

**Proof.** Consider the Lyapunov function $V_s = |\arctan(\delta)|^{\varrho_1}$. By differentiating $V_s$, one can obtain

$$\begin{aligned} \dot{V}_s &= \varrho_1|\arctan(\delta)|^{\varrho_1-1}(-2\zeta\sqrt{\rho + |\arctan(\delta)|}) \\ &\leqslant -2\zeta\varrho_1\sqrt{\rho}|\arctan(\delta)|^{\varrho_1-1} \\ &\leqslant -MV_s^{1-\frac{1}{\varrho_1}} \end{aligned} \tag{12}$$

with $M = 2\zeta\varrho_1\sqrt{\rho}$. According to Lemma 1, the nonsingular terminal sliding mode variable will converge to origin after finite time $t_4$ with $t_4 \leqslant \frac{\varrho_1}{M}|V_s(0)|^{\frac{1}{\varrho_1}}$. $\square$

**Assumption 1.** *The derivatives of the terms $d_3$ and $\bar{d}_j (j = 4, 5, 6)$ are $\dot{d}_3$ and $\dot{\bar{d}}_j$, respectively. Suppose there is a known constant $\mu$ such that $|\dot{d}_3| < \mu$ and $|\dot{\bar{d}}_j| < \mu$.*

### 3. Finite-Time ESO and DSC-Based Control Algorithm Design

*3.1. Finite-Time ESO for External Disturbances*

In practice, various disturbances impact quadrotor UAVs. These disturbances will have negative impacts on the performance of the control system, for example, by reducing the precision of trajectory tracking, increasing the chattering, and so on. In order to solve these problems, the finite-time ESO was designed to estimate the disturbances.

Define the following variables

$$x_1 = z, \, x_2 = \dot{z} \tag{13}$$

In order to design the observer and controller, one can rewrite the altitude of the position subsystem as [14]

$$\begin{cases} \dot{x}_1 = x_2 \\ \dot{x}_2 = f_1 u_F - g - \frac{k_3}{m} x_2 + d_3 \end{cases} \tag{14}$$

with $f_1 = (\cos\phi\cos\theta)/m$.

Considering the system (14), we designed an ESO to estimate $d_3$. First of all, we designed the following nonsingular terminal sliding mode variable

$$s_x = x_2 + 2\zeta\sqrt{\rho + |\arctan(x_1)|}(1 + x_1^2)\text{sign}(x_1) \tag{15}$$

Combining (14) and (15), the derivative of (15) is

$$\dot{s}_x = \dot{x}_2 + f_2$$
$$= f_1 u_F - g - \frac{k_3}{m} x_2 + d_3 + f_2 \tag{16}$$

where $f_2$ can be expressed as

$$f_2 = \begin{cases} \frac{\zeta x_2(1 + 4x_1(\arctan(x_1) - \rho))}{\sqrt{\rho - \arctan(x_1)}}, & \text{if } x_1 < 0 \\ 0, & \text{if } x_1 = 0 \\ \frac{\zeta x_2(1 + 4x_1(\arctan(x_1) + \rho))}{\sqrt{\rho + \arctan(x_1)}}, & \text{if } x_1 > 0 \end{cases} \tag{17}$$

Denote $A_1 = f_2 - g - \frac{k_3}{m} x_2$, $B_1 = f_1$, one can get

$$\dot{s}_x = A_1 + B_1 u_F + d_3 \tag{18}$$

By using the ESO technique, one can define a new state variable $Z_1 = s_x$. Meanwhile, an extended state variable $Z_2 = d_3$ can also be defined. Then, one can get

$$\begin{cases} \dot{Z}_1 = A_1 + B_1 u_F + Z_2 \\ \dot{Z}_2 = \dot{d}_3 \end{cases} \tag{19}$$

According to Assumption 1, $|\dot{Z}_2| \leqslant \mu$ is reasonable.

If we let $\hat{Z}_i$ be the observation of $Z_i$, then the observation error is $\tilde{Z}_i = \hat{Z}_i - Z_i \, (i = 1, 2)$. The finite-time ESO is designed as

$$\begin{cases} \dot{\hat{Z}}_1 = \hat{Z}_2 - r_1 \text{sig}^{b_1}(\tilde{Z}_1) - L_1 \tilde{Z}_1 + A_1 + B_1 u_F \\ \dot{\hat{Z}}_2 = -r_2 \text{sign}(\tilde{Z}_1) - L_2 \tilde{Z}_1 \end{cases} \tag{20}$$

where the gains satisfy $0 < b_1 < 1$, $r_i > 0$ and $L_i > 0 (i = 1, 2)$.

Considering the system in (19) and the ESO in (20), the error system of observer can be written as

$$
\begin{cases}
\dot{\tilde{Z}}_1 = \tilde{Z}_2 - r_1 \text{sig}^{b_1}(\tilde{Z}_1) - L_1 \tilde{Z}_1 \\
\dot{\tilde{Z}}_2 = -r_2 \text{sign}(\tilde{Z}_1) - L_2 \tilde{Z}_1 - \dot{Z}_2
\end{cases}
\tag{21}
$$

**Theorem 1.** *Considering the system in (14) and Assumption 1, the finite-time ESO is proposed as (20). The appropriate parameters $L_1$ and $L_2$ exist, which satisfy*

$$
L_1^2 \geqslant 4L_2
\tag{22}
$$

*such that observation errors $\tilde{Z} = \left[\tilde{Z}_1, \tilde{Z}_2\right]^{\text{T}}$ can converge into a small residual region in finite time.*

**Proof.** Select a Lyapunov function

$$
V_1 = \frac{1}{2}\tilde{Z}^{\text{T}}\tilde{Z}
\tag{23}
$$

From (21) and (23), $\dot{V}_1$ can be calculated as

$$
\begin{aligned}
\dot{V}_1 &= \tilde{Z}_1 \dot{\tilde{Z}}_1 + \tilde{Z}_2 \dot{\tilde{Z}}_2 \\
&= \tilde{Z}_1[\tilde{Z}_2 - r_1 \text{sig}^{b_1}(\tilde{Z}_1) - L_1 \tilde{Z}_1] + \tilde{Z}_2[-r_2 \text{sign}(\tilde{Z}_1) - \\
&\quad L_2 \tilde{Z}_1 - \dot{Z}_2] \\
&\leqslant \tilde{Z}_1 \tilde{Z}_2 - r_1|\tilde{Z}_1|^{b_1+1} - L_1 \tilde{Z}_1^2 + r_2|\tilde{Z}_2| - L_2 \tilde{Z}_1 \tilde{Z}_2 + \\
&\quad \mu|\tilde{Z}_2| \\
&\leqslant -\tilde{Z}^{\text{T}}Q\tilde{Z} + (r_2 + \mu)\|\tilde{Z}\|
\end{aligned}
\tag{24}
$$

with $Q = \begin{bmatrix} L_1 & -1 \\ L_2 & 0 \end{bmatrix}$. The characteristic equation of $Q$ can be written as

$$
\begin{aligned}
D &= |\lambda I_2 - Q| \\
&= \begin{vmatrix} \lambda - L_1 & 1 \\ -L_2 & \lambda \end{vmatrix} \\
&= \lambda^2 - L_1 \lambda + L_2
\end{aligned}
\tag{25}
$$

where $I_2$ denotes the two-dimensional identity matrix, and $\lambda$ represents a Laplase variable. The parameters $L_1$ and $L_2$ satisfy

$$
L_1^2 \geqslant 4L_2
\tag{26}
$$

such that all eigenvalues of the matrix $Q$ are positive constants. By utilizing the basic properties of matrix, the inequality

$$
\lambda_{\min}(Q)\|\tilde{Z}\|^2 \leqslant \tilde{Z}^{\text{T}}Q\tilde{Z} \leqslant \lambda_{\max}(Q)\|\tilde{Z}\|^2
\tag{27}
$$

holds, where $\lambda_{\min}(Q)$ denotes the minimum eigenvalue of $Q$, and $\lambda_{\max}(Q)$ represents the maximum eigenvalue of $Q$. Then, substituting (27) into (24) yields

$$
\begin{aligned}
\dot{V}_1 &\leqslant -\lambda_{\min}(Q)\|\tilde{Z}\|^2 + (r_2 + \mu)\|\tilde{Z}\| \\
&= [-\frac{1}{2}((\lambda_{\min}(Q))\|\tilde{Z}\| - 2(r_2 + \mu))]\|\tilde{Z}\| - \frac{1}{2}\lambda_{\min}(Q)\|\tilde{Z}\|^2
\end{aligned}
\tag{28}
$$

If the condition $\|\tilde{Z}\| \geqslant 2(r_2 + \mu)/\lambda_{\min}(Q)$ is satisfied, one can get $\dot{V}_1 \leqslant 0$. When $\|\tilde{Z}\| \geqslant 2(r_2 + \mu)/(\theta_1 \lambda_{\min}(Q))$ with $0 < \theta_1 < 1$, one can obtain

$$\dot{V}_1 \leqslant -M_1 V_1^{\frac{1}{2}} - M_2 V_1 \tag{29}$$

with $M_1 = \min \frac{\sqrt{2}(r_2 + \mu)(1 - \theta_1)}{\theta_1}$, $M_2 = \lambda_{\min}(Q)$.

Therefore, the observation error $\tilde{Z}$ will converge into the set $\{\tilde{Z}|\|\tilde{Z}\| \leqslant 2(r_2 + \mu)/\lambda_{\min}(Q)\}$ after finite-time $t_{f1}$. According to Lemma 2, the time $t_{f1}$ is given by

$$t_{f1} \leqslant \frac{2}{M_2} \ln \frac{M_2\sqrt{V_1(0)} + M_1}{M_1} \tag{30}$$

The proof is completed. □

**Remark 1.** *The ESO in* (20) *is comprised of two parts: a linear part and a nonlinear part. The ESO has a fast response for the linear part with a small damping ratio. When the observation error tends to zero, the nonlinear linear part increases the damping ratio and reduces the overshoot caused by the linear part. Although the nonlinear part has no effect on the proof process, it has an important impact on the performance of ESO* [39]. *It can be seen for the ESO* (20) *that the composite nonlinear ESO* [39] *is uniformly ultimately bounded, while the ESO* (20) *presented in this paper can guarantee the practically finite-time stability of the error system.*

### 3.2. NTSMC and DSC-Based Finite-Time Position Controller Design

In this section, a NTSMC and a DSC-based control scheme are used to design the finite-time position tracking controller. At the same time, this scheme is applied to the attitude subsystem. In order to decrease the negative impacts of the ESO error on the stability of the position subsystem, we introduce an adaptive law to compensate for the observation error.

**Step1:** Define the dynamic errors $e_{z1} = x_1 - x_{1d}$ and $e_{z2} = x_2 - \beta_f$, where $x_{1d}$ denotes the desired altitude and $\beta_f$ is the output of finite-time filter. Construct the following virtual control law

$$\beta = \dot{x}_{1d} - l_1 |e_{z1}|^{c_1} \text{sign}(e_{z1}) \tag{31}$$

with $l_1 > 0, 0 < c_1 < 1$.

To address the problem of "explosion of complexity" in the traditional backstepping design, the finite-time filter is introduced as

$$\begin{cases} \dot{\beta}_f = \varphi_1 - \lambda_1 |\beta_f - \beta|^{p_1} \text{sign}(\beta_f - \beta) \\ \dot{\varphi}_1 = -\lambda_2 |\beta_f - \beta|^{p_2} \text{sign}(\beta_f - \beta) \end{cases} \tag{32}$$

with $0 < p_1 < 1, p_2 = 2p_1 - 1$. $\lambda_1$ and $\lambda_2$ are positive constants. From [44], it is known that $\beta_f$ converges to $\beta$ after finite time, namely $|\beta_f - \beta| = 0$ after finite time.

Considering a Lyapunov function $V_2 = 0.5e_{z1}^2$, taking the time derivative of $V_2$ yields

$$\begin{aligned} \dot{V}_2 &= e_{z1}\dot{e}_{z1} \\ &= e_{z1}(e_{z2} + \beta_f - \dot{x}_{1d}) \end{aligned} \tag{33}$$

When $\beta_f$ converges to $\beta$, by substituting (31) into (33), one can obtain

$$\dot{V}_2 \leqslant -l_1 |e_{z1}|^{c_1+1} + e_{z1}e_{z2} \tag{34}$$

**Step2:** Design the following nonsingular terminal sliding mode variable

$$s_1 = e_{z2} + 2\zeta\sqrt{\rho + |\arctan(e_{z1})|}(1 + e_{z1}^2)\text{sign}(e_{z1}) \tag{35}$$

The proposed position control law is

$$u_F = -\frac{1}{f_1}\left[-g - \frac{k_3}{m}x_2 + \hat{Z}_2 - \dot{\beta}_f + f_2' + \varepsilon|s_1|^{c_1}\text{sign}(s_1) + e_{z1} + \hat{a}_1\text{sign}(s_1)\right] \tag{36}$$

where $\varepsilon$ is a positive constant, $\hat{a}_1$ is the estimation of $a_1$, and $a_1$ is the upper bound of the observation error $\tilde{Z}_2$. $\hat{a}_1$ is defined as

$$\dot{\hat{a}}_1 = m_1|s_1| - n_1\hat{a}_1 \tag{37}$$

with $\hat{a}_1(0) \geqslant 0$, $m_1 > 0$ and $n_1 > 0$. According to Lemma 2 in [45], one can get $0 < \hat{a}_1 < \bar{a}_1$, where $\bar{a}_1$ is positive scalar. The function $f_2'$ is given by

$$f_2' = \begin{cases} \frac{\zeta\dot{e}_{z1}(1+4e_{z1}(\arctan(e_{z1})-\rho))}{\sqrt{\rho-\arctan(e_{z1})}}, & \text{if } e_{z1} < 0 \\ 0, & \text{if } e_{z1} = 0 \\ \frac{\zeta\dot{e}_{z1}(1+4e_{z1}(\arctan(e_{z1})+\rho))}{\sqrt{\rho+\arctan(e_{z1})}}, & \text{if } e_{z1} > 0 \end{cases} \tag{38}$$

**Theorem 2.** *Consider the position subsystem presented in* (14) *and finite-time ESO in* (20). *If the NTSMC-based finite-time dynamic surface controller is designed as* (36), *then the states of system can be practically finite-time stabilized.*

**Proof.** Consider a Lyapunov function $V_3 = V_2 + 0.5s_1^2 + (\hat{a}_1 - a_1)^2/2m_1$. By differentiating $V_3$, one can obtain

$$\dot{V}_3 = \dot{V}_2 + s_1\dot{s}_1 + \frac{1}{m_1}(\hat{a}_1 - a_1)\dot{\hat{a}}_1 \tag{39}$$

Considering the control law presented in (36), the derivative of the nonsingular terminal sliding mode manifold is

$$\begin{aligned} \dot{s}_1 &= \dot{e}_{z2} + f_2' \\ &= -\varepsilon|s_1|^{c_1}\text{sign}(s_1) - e_{z1} - \hat{a}_1\text{sign}(s_1) - \tilde{Z}_2 \end{aligned} \tag{40}$$

By substituting (34) and (40) into (39), one can obtain

$$\begin{aligned} \dot{V}_3 &= \dot{V}_2 + s_1[-\varepsilon|s_1|^{c_1}\text{sign}(s_1) - e_{z1} - \hat{a}_1\text{sign}(s_1) - \tilde{Z}_2] + \frac{1}{m_1}(\hat{a}_1 - a_1)\dot{\hat{a}}_1 \\ &= \dot{V}_2 + s_1[-\varepsilon|s_1|^{c_1}\text{sign}(s_1) - e_{z1} - \hat{a}_1\text{sign}(s_1) - \tilde{Z}_2] + (\hat{a}_1 - a_1)|s_1| \\ &\quad - \frac{n_1}{m_1}(\hat{a}_1 - a_1)\hat{a}_1 \\ &\leqslant -l_1|e_{z1}|^{c_1+1} + e_{z1}e_{z2} - \varepsilon|s_1|^{c_1+1} - s_1e_{z1} + |s_1||\tilde{Z}_2| - \hat{a}_1|s_1| \\ &\quad + (\hat{a}_1 - a_1)|s_1| - \frac{n_1}{m_1}(\hat{a}_1 - a_1)\hat{a}_1 \\ &\leqslant -l_1|e_{z1}|^{c_1+1} - \varepsilon|s_1|^{c_1+1} - \frac{n_1}{m_1}(\hat{a}_1 - a_1)\hat{a}_1 \\ &= -l_1|e_{z1}|^{c_1+1} - \varepsilon|s_1|^{c_1+1} - \frac{n_1}{m_1}|\hat{a}_1 - a_1|^{c_1+1} + \frac{n_1}{m_1}|\hat{a}_1 - a_1|^{c_1+1} - \frac{n_1}{m_1}(\hat{a}_1 - a_1)\hat{a}_1 \end{aligned} \tag{41}$$

The following inequality holds because $\hat{a}_1 > 0$.

$$\begin{aligned} &\frac{n_1}{m_1}|\hat{a}_1 - a_1|^{c_1+1} - \frac{n_1}{m_1}(\hat{a}_1 - a_1)\hat{a}_1 \\ &= \frac{n_1}{m_1}[|\hat{a}_1 - a_1|^{c_1+1} - (\hat{a}_1 - a_1)^2 - (\hat{a}_1 - a_1)a_1] \\ &\leqslant \frac{n_1}{m_1}[|\hat{a}_1 - a_1|^{c_1+1} - |\hat{a}_1 - a_1|^2] + \frac{n_1}{m_1}a_1^2 \end{aligned} \tag{42}$$

(1) When $0 < |\hat{a}_1 - a_1| < 1$, one can get

$$0 \leqslant |\hat{a}_1 - a_1|^{c_1+1} - |\hat{a}_1 - a_1|^2 \leqslant \varsigma \tag{43}$$

with $\varsigma = b^{\frac{b}{1-b}} - b^{\frac{1}{1-b}}$, $b = \frac{c_1+1}{2}$.

(2) When $|\hat{a}_1 - a_1| > 1$, the inequality

$$|\hat{a}_1 - a_1|^{c_1+1} - |\hat{a}_1 - a_1|^2 < 0 \tag{44}$$

holds.

Hence, it is apparent that the inequality $|\hat{a}_1 - a_1|^{c_1+1} - |\hat{a}_1 - a_1|^2 \leqslant \varsigma$ is guaranteed in any case. Further, one can obtain

$$\frac{n_1}{m_1}|\hat{a}_1 - a_1|^{c_1+1} - \frac{n_1}{m_1}(\hat{a}_1 - a_1)\hat{a}_1 \leqslant \frac{n_1}{m_1}(\varsigma + a_1^2) \tag{45}$$

Therefore, $\dot{V}_3$ can be expressed as

$$\begin{aligned}
\dot{V}_3 &\leqslant -l_1|e_{z1}|^{c_1+1} - \varepsilon|s_1|^{c_1+1} - \frac{n_1}{m_1}|\hat{a}_1 - a_1|^{c_1+1} + \frac{n_1}{m_1}(\varsigma + a_1^2) \\
&\leqslant -M_3 V_3^{\frac{c_1+1}{2}} + \frac{n_1}{m_1}(\varsigma + a_1^2)
\end{aligned} \tag{46}$$

with $M_3 = 2^{\frac{c_1+1}{2}} \min\{l_1, \varepsilon, n_1 m_1^{\frac{c_1-1}{2}}\}$ and $n_1(\varsigma + a_1^2)/m_1 > 0$.

It can be seen from Lemma 3 that the system in (14) is practically finite-time stable. $\square$

**Remark 2.** *When $s_1(t) = 0$ is satisfied, one can obtain*

$$e_{z2} = -2\zeta\sqrt{\rho + |\arctan(e_{z1})|}(1 + e_{z1}^2)\text{sign}(e_{z1}) \tag{47}$$

*Consider the derivative of $V_2$, the inequality*

$$\dot{V}_2 \leqslant -l_1|e_{z1}|^{c_1+1} - F_1|e_{z1}| \tag{48}$$

*holds with $F_1 = 2\zeta\sqrt{\rho + |\arctan(e_{z1})|}(1 + e_{z1}^2) > 0$. Then, one can get*

$$\dot{V}_2 \leqslant -2^{\frac{c_1+1}{2}}l_1 V_2^{\frac{c_1+1}{2}} \tag{49}$$

*According to Lemma 1, the nonsingular terminal sliding mode variable $s_1$ is finite-time stable.*

**Remark 3.** *The traditional DSC [20] introduces a first-order low-pass filter to address the "explosion of complexity", which can only achieve the uniform ultimate boundedness of the control system. A filter (32) is introduced to ensure the finite-time stability of the system. Meanwhile, the estimation error of filter can converge to origin after finite time. Therefore, the accuracy and convergence rate of the control system are improved.*

### 3.3. Finite-Time ESO for External Disturbances and Parameter Uncertainties

Denote $\iota = [\iota_1, \iota_2, \iota_3]^{\text{T}} = [\phi, \theta, \psi]^{\text{T}}$, $y_{1i} = \iota_i$, $y_{2i} = \dot{\iota}_i$ ($i = 1, 2, 3$). The new variables $f_{3i} = J_i^{-1}$ and $\bar{d}_j = f_{3i}(d_j - \Delta J_i \dot{y}_{2i})$ ($i = 1, 2, 3; j = 4, 5, 6$) are defined. Then, the attitude dynamics in (1) can be rewritten as

$$\begin{cases} \dot{y}_{1i} = y_{2i} \\ \dot{y}_{2i} = -f_{3i}k_j\xi_1 y_{2i} + f_{3i}\xi_2 u_i + \bar{d}_j \end{cases} \tag{50}$$

with $\xi_1 = \begin{cases} l, & \text{if } i = 1,2 \\ 1, & \text{if } i = 3 \end{cases}, \xi_2 = \begin{cases} l, & \text{if } i = 1,2 \\ c, & \text{if } i = 3 \end{cases}$ .

Firstly, the nonsingular terminal sliding mode manifold is as follows

$$s_{yi} = y_{2i} + 2\zeta\sqrt{\rho + |\arctan(y_{1i})|}(1 + y_{1i}^2)\text{sign}(y_{1i}) \tag{51}$$

with $i = 1, 2, 3$. Taking the derivative of $s_{yi}$, one gets

$$\begin{aligned} \dot{s}_{yi} &= \dot{y}_{2i} + f_{2i} \\ &= -f_{3i}k_j\xi_1 y_{2i} + f_{3i}\xi_2 u_i + \bar{d}_j + f_{2i} \end{aligned} \tag{52}$$

where $f_{2i}(i = 1, 2, 3)$ is represented as

$$f_{2i} = \begin{cases} \frac{\zeta y_{2i}(1+4y_{1i}(\arctan(y_{1i})-\rho))}{\sqrt{\rho-\arctan(y_{1i})}}, & \text{if } y_{1i} < 0 \\ 0, & \text{if } y_{1i} = 0 \\ \frac{\zeta y_{2i}(1+4y_{1i}(\arctan(y_{1i})+\rho))}{\sqrt{\rho+\arctan(y_{1i})}}, & \text{if } y_{1i} > 0 \end{cases} \tag{53}$$

Then, (52) can be rewritten as

$$\dot{s}_{yi} = A_{1i} + B_{1i}u_i + \bar{d}_j \tag{54}$$

with $A_{1i} = f_{2i} - f_{3i}k_j\xi_1 y_{2i}$, $B_{1i} = f_{3i}\xi_2 (i = 1, 2, 3)$.

By utilizing the ESO technique, one can define two new variables $Z_{3i} = s_{yi}$ and $Z_{4i} = \bar{d}_j$. According to Assumption 1, one can get $|\dot{Z}_{4i}| \leqslant \mu$. The system in (54) can be expressed as

$$\begin{cases} \dot{Z}_{3i} = A_{1i} + B_{1i}u_i + Z_{4i} \\ \dot{Z}_{4i} = \dot{\bar{d}}_j \end{cases} \tag{55}$$

with $i = 1, 2, 3$. The observations of $Z_{3i}$ and $Z_{4i}$ are $\hat{Z}_{3i}$ and $\hat{Z}_{4i}$. Then, the observation errors of $Z_{3i}$ and $Z_{4i}$ are $\tilde{Z}_{3i} = \hat{Z}_{3i} - Z_{3i}$ and $\tilde{Z}_{4i} = \hat{Z}_{4i} - Z_{4i}$, respectively. A finite-time ESO is designed as

$$\begin{cases} \dot{\hat{Z}}_{3i} = \hat{Z}_{4i} - r_{3i}\text{sig}^{b_3}(\tilde{Z}_{3i}) - L_{3i}\tilde{Z}_{3i} + A_{1i} + B_{1i}u_i \\ \dot{\hat{Z}}_{4i} = -r_{4i}\text{sign}(\tilde{Z}_{3i}) - L_{4i}\tilde{Z}_{3i} \end{cases} \tag{56}$$

with $0 < b_3 < 1$, $r_{3i} > 0$, $r_{4i} > 0$, $L_{3i} > 0$ and $L_{4i} > 0 (i = 1, 2, 3)$.

Considering (55) and (56), the observation error dynamics can be written as

$$\begin{cases} \dot{\tilde{Z}}_{3i} = \tilde{Z}_{4i} - r_{3i}\text{sig}^{b_3}(\tilde{Z}_{3i}) - L_{3i}(\tilde{Z}_{3i}) \\ \dot{\tilde{Z}}_{4i} = -r_{4i}\text{sign}(\tilde{Z}_{3i}) - L_{4i}(\tilde{Z}_{3i}) - \dot{Z}_{4i} \end{cases} \tag{57}$$

**Theorem 3.** *Considering the attitude subsystem presented in* (50) *and Assumption 1, the proposed finite-time ESO is* (56)*. The proper parameters $L_{3i}$ and $L_{4i}$ are chosen, which satisfy*

$$L_{3i}^2 > 4L_{4i} \tag{58}$$

*such that observation errors $\tilde{\mathbf{Z}}_{yi} = \begin{bmatrix} \tilde{Z}_{3i}, & \tilde{Z}_{4i} \end{bmatrix}^{\mathrm{T}}(i = 1, 2, 3)$ can converge into a small residual region in finite time.*

**Proof.** The proof is omitted to save space, as it is similar to Theorem 1. Eventually, the estimation error $\tilde{\mathbf{Z}}_{yi}$ will converge to the neighborhood of the origin after finite time. $\square$

*3.4. NTSMC and DSC-Based Finite-Time Attitude Controller Design*

Considering the system in (50), a controller based on the NTSMC and DSC techniques was designed.

**Step1:** Define the following auxiliary variables

$$e_{t1i} = y_{1i} - y_{1di}, e_{t2i} = y_{2i} - \alpha_{fi} \tag{59}$$

where $y_{1di}(i = 1, 2, 3)$ denotes the desired attitude and $\alpha_{fi}$ is the output of the finite-time filter. Construct the following virtual control law

$$\alpha_i = \dot{y}_{1di} - l_{2i}|e_{t1i}|^{c_{2i}}\text{sign}(e_{t1i}) \tag{60}$$

with $l_{2i} > 0, 0 < c_{2i} < 1(i = 1, 2, 3)$.

To solve the problem of the "explosion of complexity" in the traditional backstepping design, the finite-time filter is introduced

$$\begin{cases} \dot{\alpha}_{fi} = \varphi_{2i} - \lambda_{3i}|\alpha_{fi} - \alpha_i|^{p_{3i}}\text{sign}(\alpha_{fi} - \alpha_i) \\ \dot{\varphi}_{2i} = -\lambda_{4i}|\alpha_{fi} - \alpha_i|^{p_{4i}}\text{sign}(\alpha_{fi} - \alpha_i) \end{cases} \tag{61}$$

with $0 < p_{3i} < 1$, $p_{4i} = 2p_{3i} - 1$, $\lambda_{3i} > 0$, $\lambda_{4i} > 0(i = 1, 2, 3)$. As shown in [44], $\alpha_{fi}$ will converge to $\alpha_i$ in finite time, namely $|\alpha_{fi} - \alpha_i| = 0$ after finite time.

Consider the Lyapunov function $V_{4i} = 0.5e_{t1i}^2$ with $i = 1, 2, 3$. The proof is similar to (33), and one can obtain

$$\dot{V}_{4i} \leqslant -l_{2i}|e_{t1i}|^{c_{2i}+1} + e_{t1i}e_{t2i} \tag{62}$$

**Step2:** Design the nonsingular terminal sliding mode variable as

$$s_{2i} = e_{t2i} + 2\zeta\sqrt{\rho + |\arctan(e_{t1i})|}(1 + e_{t1i}^2)\text{sign}(e_{t1i}) \tag{63}$$

with $i = 1, 2, 3$. According to Remark 2, the finite-time stability of the nonsingular terminal sliding mode variable $s_{2i}$ is guaranteed.

The attitude controller is designed as

$$u_i = -\frac{1}{f_{3i}\xi_2}[-f_{3i}k_j\xi_1 y_{2i} + \hat{Z}_{4i} - \dot{\alpha}_{fi} + f'_{2i} + \varepsilon_{2i}|s_{2i}|^{c_{2i}}\text{sign}(s_{2i}) + e_{t1i} + \hat{a}_{2i}\text{sign}(s_{2i})] \tag{64}$$

with $\varepsilon_{2i} > 0$ and $i = 1, 2, 3$. $\hat{a}_{2i}$ is the estimation of $a_{2i}$, $a_{2i}$ is the upper bound of the estimation error $\tilde{Z}_{4i}$, $\hat{a}_{2i}$ is developed as

$$\dot{\hat{a}}_{2i} = m_{2i}|s_{2i}| - n_{2i}\hat{a}_{2i} \tag{65}$$

with $\hat{a}_{2i}(0) \geqslant 0$, $m_{2i} > 0$ and $n_{2i} > 0$. According to Lemma 2 in [45], one can get $0 < \hat{a}_{2i} < \bar{a}_{2i}$, where $\bar{a}_{2i}$ is positive scalar. $f'_{2i}$ can be expressed as

$$f'_{2i} = \begin{cases} \frac{\zeta\dot{e}_{t1i}(1+4e_{t1i}(\arctan(e_{t1i})-\rho))}{\sqrt{\rho-\arctan(e_{t1i})}}, & \text{if } e_{t1i} < 0 \\ 0, & \text{if } e_{t1i} = 0 \\ \frac{\zeta\dot{e}_{t1i}(1+4e_{t1i}(\arctan(e_{t1i})+\rho))}{\sqrt{\rho+\arctan(e_{t1i})}}, & \text{if } e_{t1i} > 0 \end{cases} \tag{66}$$

**Theorem 4.** *Consider the attitude subsystem presented in (50). If the NTSMC-based finite-time dynamic surface controllers is designed as (64), then the states of the system can be practically finite-time stabilized.*

**Proof.** Choose a Lyapunov function

$$V_{5i} = V_{4i} + \frac{1}{2}s_{2i}^T s_{2i} + \frac{1}{2m_{2i}}(\hat{a}_{2i} - a_{2i})^2 \tag{67}$$

with $i = 1, 2, 3$. The proof process of Theorem 4 is similar to Theorem 2, and the attitude tracking control system will also converge into a small residual region after finite time. In order to save space, the certification step is omitted. □

**Remark 4.** *The second Lyapunov method is used to judge the stability of the system. That is, a positive definite scalar function $V(x)$ is defined as an imaginary generalized energy function, and then the stability of the system is judged according to the symbolic characteristics of $\dot{V}(x)$. According to Lemmas 1–3, the finite-time stability of the system is proved in this paper.*

## 4. Simulation and Analysis

### 4.1. Numerical Simulation Results

In this section, numerical simulations are applied to validate the effectiveness of the presented control algorithm. The quadrotor UAV model parameters are described as $m = 1.1\,\text{kg}$, $k_i(i = 1, 2, 3) = 0.01\,\text{Ns/m}$, $J_i(i = 1, 2) = 1.22\,\text{kg}\cdot\text{m}^2$, $g = 9.81\,\text{m/s}^2$, $l = 1\,\text{m}$, $c = 1.5$, $k_i(i = 4, 5, 6) = 0.012\,\text{Ns/m}$ and $J_3 = 2.2\,\text{kg}\cdot\text{m}^2$. It is assumed that the external disturbances and parameter uncertainties are respectively given as $\boldsymbol{d} = \begin{bmatrix} 0.1\cos(0.2t)\text{N}, & 0.5\sin(0.1t)\text{N.m}, & 0.6\cos(0.5t)\text{N.m}, & 0.2\sin(0.2t)\text{N.m} \end{bmatrix}^\text{T}$, and $\boldsymbol{\Delta}(\boldsymbol{J}) = \begin{bmatrix} 0.1\cos(0.2t)\text{kg}\cdot\text{m}^2, & 0.2\sin(0.3t)\text{kg}\cdot\text{m}^2, & 0.2\cos(0.5t)\text{kg}\cdot\text{m}^2 \end{bmatrix}^\text{T}$.

The parameters of ESO presented in (20) are $\zeta = 0.0001$, $\rho = 0.0001$, $r_1 = 2$, $r_2 = 0.02$, $b_1 = 0.9$, $L_1 = 1$, $L_2 = 0.2$. The gains of position control law are chosen as $l_1 = 1$, $c_1 = 0.8$, $\lambda_1 = 0.2$, $\lambda_2 = 0.3$, $p_1 = 0.6$, $p_2 = 0.2$, $\varepsilon = 0.5$, $m_1 = 0.1$, and $n_1 = 0.1$. The main parameters of the attitude subsystem are $r_{31} = 2$, $r_{32} = 1$, $r_{33} = 0.6$, $r_{41} = 0.05$, $r_{42} = 0.5$, $r_{43} = 0.2$, $b_3 = 0.8$, $\varepsilon_{21} = 2$, $\varepsilon_{22} = 5$, $\varepsilon_{23} = 1$, $c_{21} = 0.9$, $c_{22} = 0.9$, and $c_{23} = 0.8$.

To illustrate the superiority of the proposed control scheme, a finite-time dynamic surface control (FTDSC) scheme [14] is introduced. External disturbances and parameter uncertainties are restricted to the same values in the proposed algorithm and FTDSC to make a fair comparison. Figure 2 shows a comparison of trajectory/attitude tracking under the proposed scheme and the FTDSC scheme. It is observed that the proposed scheme obtains a better performance in terms of tracking the desired trajectory/attitude. Figure 3 compares the tracking errors. It is shown that the proposed scheme has better convergence performance. A comparison of the linear/angular velocity is shown in Figure 4. It can be concluded that the proposed scheme provides better stability. Figure 5 illustrates the control inputs under the proposed scheme and the FTDSC scheme. It can be clearly seen that the control inputs of the proposed scheme are appropriate. Figure 6 describes the convergence performance of the observation errors of the ESOs. It is obvious that the proposed ESOs can estimate the actual disturbances successfully with a settling time of less than 10 s. Therefore, highly precise tracking control can be accomplished via the proposed scheme.

The root-mean-square error (RMSE) and the mean absolute error (MAE) were used as performance indicators to assess the results of the comparison simulation. The results are listed in Table 1. Overall, the proposed scheme performs better than the FTDSC. Although the RMSE of the tracking errors of $\phi$ and $\theta$ in the proposed scheme are larger than that of FTDSC, the tracking errors of $\phi$ and $\theta$ in the proposed scheme have better stability.

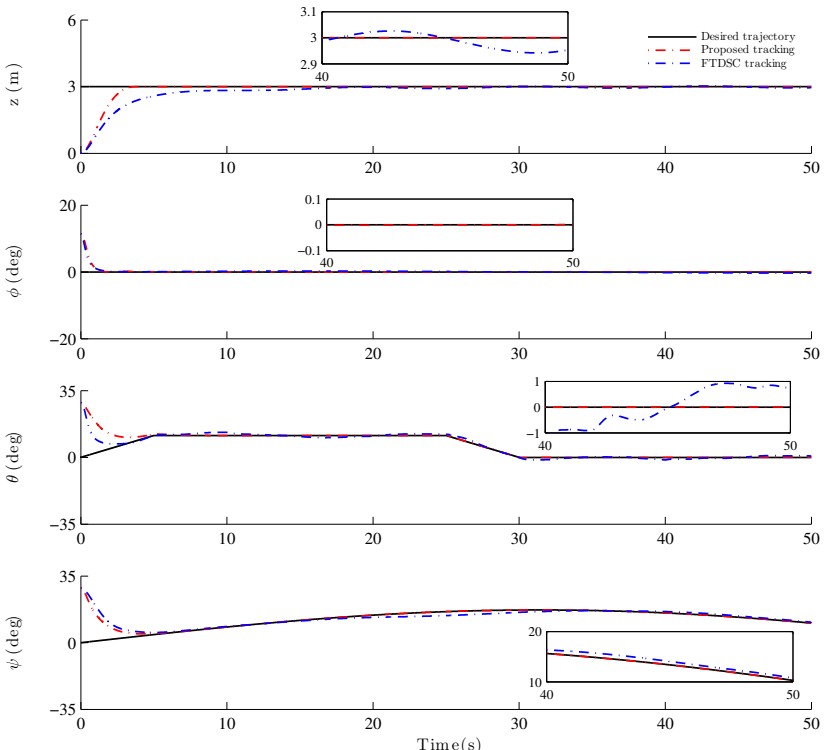

**Figure 2.** Time response of trajectory/attitude tracking.

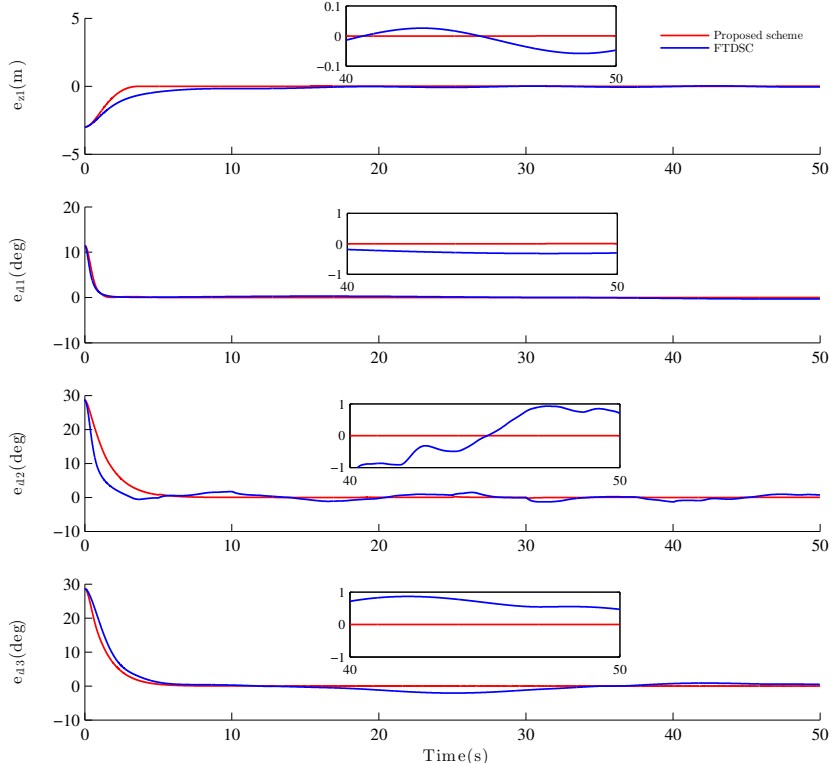

**Figure 3.** Time response of trajectory/attitude tracking errors.

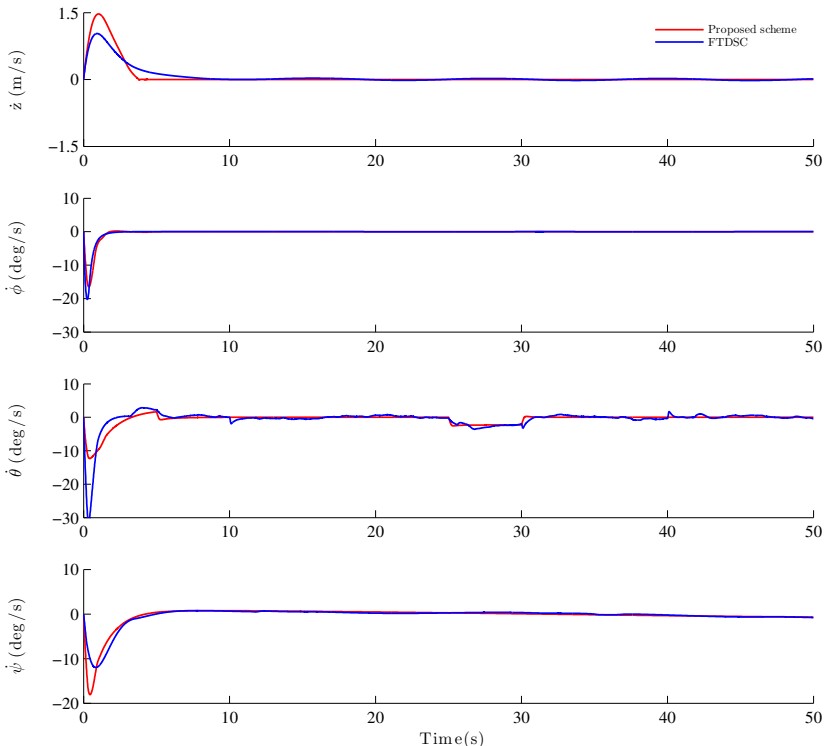

**Figure 4.** Time response of linear/angular velocity.

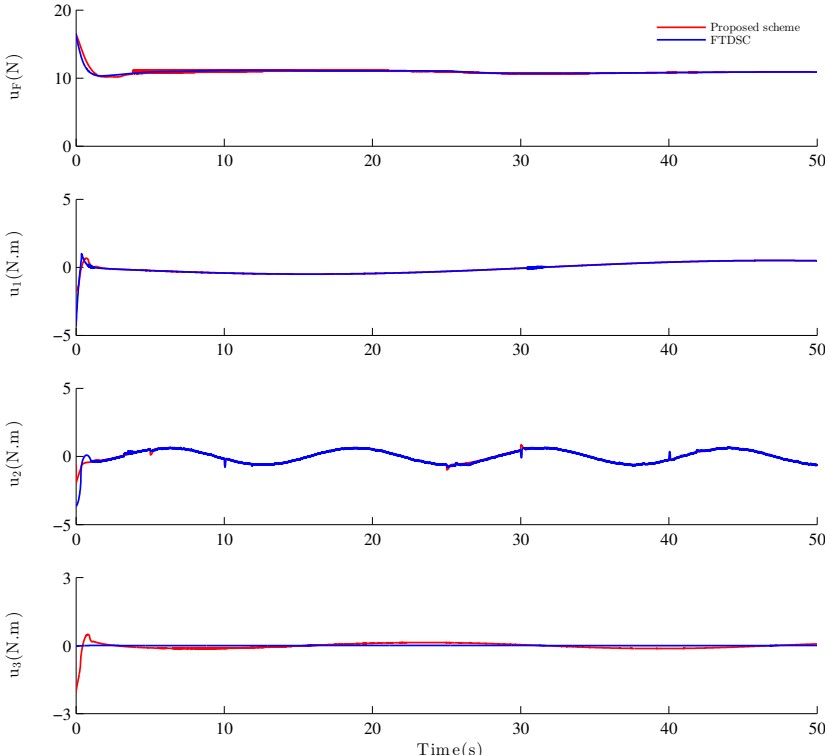

**Figure 5.** Time response of control inputs.

**Table 1.** Trajectory tracking performance evaluation.

| Index | Control Schemes | $z$ (m) | $\phi$ (deg) | $\theta$ (deg) | $\psi$ (deg) |
|-------|-----------------|---------|--------------|----------------|--------------|
| MAE   | Proposed scheme | 0.0870  | 0.1322       | 0.9053         | 0.7940       |
|       | FTDSC           | 0.2032  | 0.2789       | 1.0750         | 1.6784       |
| RMSE  | Proposed scheme | 0.4258  | 0.9800       | 3.8043         | 3.5587       |
|       | FTDSC           | 0.5119  | 0.8936       | 2.8448         | 4.2335       |

**Remark 5.** *The parameters of the proposed control strategy affect the settling time and convergence accuracy of the control system. According to Lyapunov's theory, some parameters have a range of values, and other parameters are obtained by a trial and error approach.*

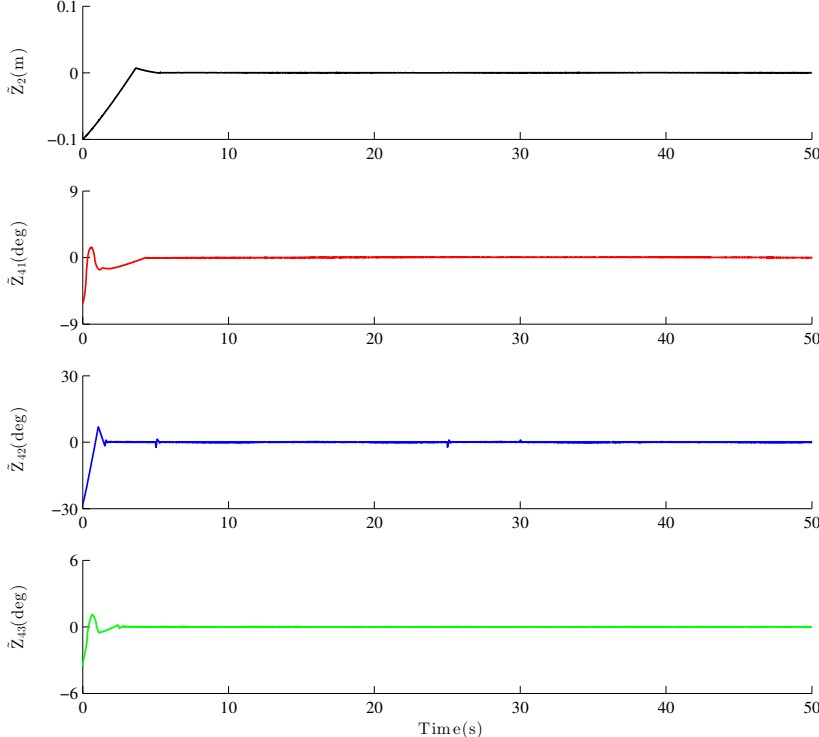

**Figure 6.** Time response of observation errors of ESO.

*4.2. Monte Carlo Results*

This subsection describes a Monte Carlo simulation with 50 runs that was carried out for attitude control to further verify the effectiveness of the proposed algorithm. The random external disturbances and random parameter uncertainties are added in the form of

$$d = \begin{bmatrix} 0.1\cos(\text{rand} * 0.2t)\text{N} \\ \text{rand} * 0.2\sin(\text{rand} * 0.1t)\text{N.m} \\ \text{rand} * 0.1\cos(\text{rand} * 0.1t)\text{N.m} \\ \text{rand} * 0.2\sin(\text{rand} * 0.2t)\text{N.m} \end{bmatrix} \tag{68}$$

$$\Delta(J) = \begin{bmatrix} \text{rand} * 0.1\cos(\text{rand} * 0.2t) \\ \text{rand} * 0.1\sin(\text{rand} * 0.1t) \\ \text{rand} * 0.1\cos(\text{rand} * 0.5t) \end{bmatrix} \text{kg} \cdot \text{m}^2 \tag{69}$$

The tracking errors are shown in Figure 7, and it can be seen that the tracking errors converge to zero in finite time. In other words, the desired attitude commands can be tracked by the proposed control scheme. The observation errors are depicted in Figure 8. Tt can be seen that the proposed ESO can achieve a satisfactory performance, even under exposure to random external disturbances and random parameter uncertainties.

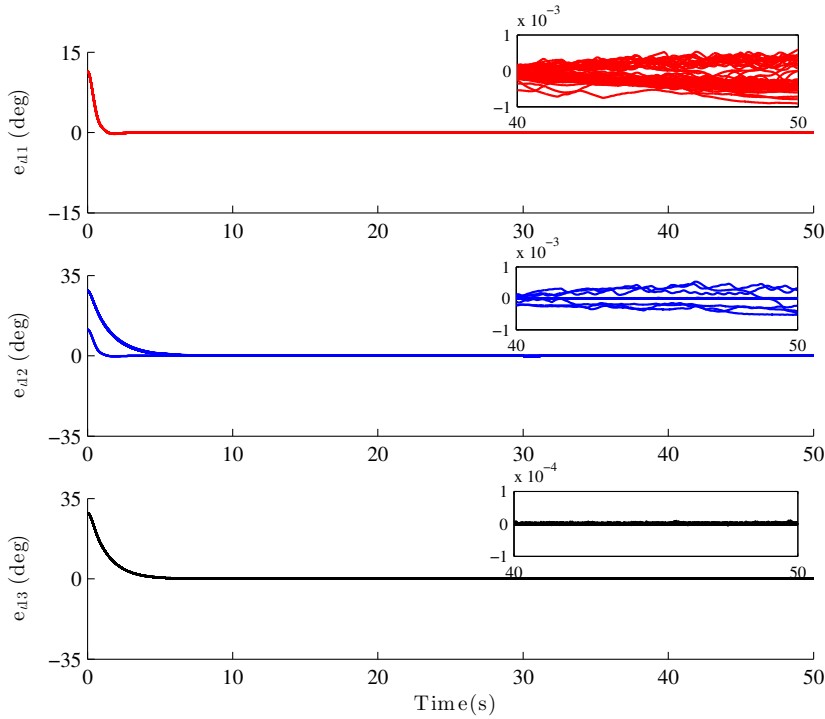

**Figure 7.** Time response of the tracking errors in the MC simulation.

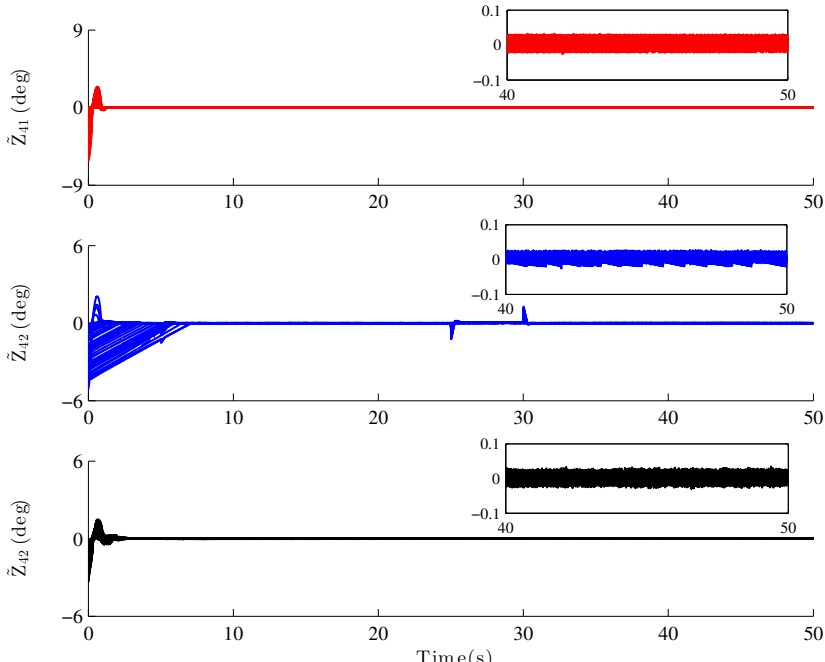

**Figure 8.** Time response of the observation errors in the MC simulation.

## 5. Conclusions

This work addresses the problem of finite-time trajectory tracking of quadrotor UAVs in the presence of external disturbances and parameter uncertainties. Two NTSMC technique-based finite-time ESOs for position and attitude subsystems are developed to estimate the external disturbances and/or the parameter uncertainties caused by wind disturbances. Based on the DSC and NTSMC techniques, two trajectory tracking controllers, which ensure that the tracking errors can converge to a small residual region after finite time, are presented. Finally, numerical simulation results show the satisfactory performance of the proposed control strategy. However, the proposed control scheme is

capable of handling the disturbances under Assumption 1. This is also a problem that should be addressed in future work.

**Author Contributions:** Conceptualization, Y.N. and H.B.; methodology, Y.N. and H.B.; software, Y.N. and H.Z.; validation, Y.N., H.Z. and W.G.; formal analysis, Y.N., H.Z. and H.B.; data curation, Y.N.; writing—original draft preparation, Y.N.; writing—review and editing, Y.N. and W.G.; visualization, Y.N.; supervision, W.G. and F.Y.; project administration, F.Y.; funding acquisition, F.Y. All authors have read and agreed to the published version of the manuscript.

**Funding:** This research was funded by the Shanghai Science and Technology Committee under grant number 20dz1203005, and the National Natural Science Foundation of China under grant number 61703272, and the Project funded by China Postdoctoral Science Foundation under grant number 2019M661467.

**Institutional Review Board Statement:** Not applicable.

**Informed Consent Statement:** Not applicable.

**Data Availability Statement:** Not applicable.

**Conflicts of Interest:** The authors declare no conflict of interest.

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
