# Peer review of "Nonsingular Terminal Sliding Mode Based Finite-Time Dynamic Surface Control for a Quadrotor UAV"

_algorithms, doi:10.3390/a14110315_

Round 1

Reviewer 1 Report

This paper aims to present a nonsingular terminal sliding mode-based finite-time dynamic surface control for a quadrotor subject to external disturbances and parameter uncertainties. It develops an extended state observer based on the nonsingular terminal sliding mode variable to estimate the external position of the UAV.  Then the dynamic surface controller is proposed. The same strategy is done for the attitude subsystem. In my opinion, the topic is interesting and the results seem correct. However, the paper must be improved.

*The language is poor. It must be improved considerably. There are a lot of typos and language issues. 

*The literature review misses one important type of observers that are being recently used in this scenarios, e.g. filtered observers, see for instance: Filtered Observer-Based IDA-PBC Control for Trajectory Tracking of a Quadrotor, IEEE access; Filtered Output Feedback Tracking Control of a Quadrotor UAV, IFAC-PapersOnLine.  

*Please ensure you define all the acronyms in first use, even if they are already defined in the abstract, e.g. ESO, NTSMC.

*There are some abrupt changes between paragraphs in the introduction, e.g. line 44-45

*Use the standard letter to denote real numbers and in general sets: \mathbb{R}, \mathbb{U}.

*Be consistent in your nomenclature and in the presentation of the paper. In general most of the paper omits the time dependency of the variables, however, in some equations the variables show their time dependency. 

* In equation 1, x denotes position. However, when defining the lemmas, x denotes a general state vector. Check the definition of the variables thoroughly.

* Assumption 1. Says that \bar{d}_j is defined below, however, is not presented. 

* One numerical test is not enough. Please increase the numerical tests. For the disturbance test please include some random disturbances, the ones presented are very smooth. 

* Include performance indexes to assess the performance. 

* Improve the conclusion section and define clearly the limitations of the approach.

Reviewer 2 Report

In this paper, the nonsingular finite time sliding mode combined with dynamic surface control and extended state observer for UAV was studied. Simulation results to show the effectiveness of the proposed methods.

Questions for this paper are as follows:

The presented finite time sliding mode control, dynamic surface control, and extended state observer method have been studied by many researchers. Then, the proposed method seems to be mixed version of these methods.

What is the contribution of the authors for this paper?

2. Compare the performance of the proposed finite time sliding surface in (15) with respect to conventional finite time sliding mode surface.

3. Simulation trajectory for USV is simple. Further real flight trajectory simulation results are required.

Reviewer 3 Report

The organization and content of this paper is suitable and mathematical proofs are clear. Though, I think that this paper should be revised carefully. I suggest to consider the following comments to improve the quality of the paper:

  • The novelty of the proposed method should be highlighted carefully. Moreover, the importance of the suggested method should be further addressed. 
  • No issue regarding complexity of the proposed method has been presented.
  • The paper should be reviewed carefully, in order to correct all the typing, grammar and English errors.
  • The authors should give a Remark to illustrate how the design parameters effect the control performance, and how to choose these parameters.
  • The most recent related works to finite time control and UAV systems should be cited in the literature review such as the following cases: https://ieeexplore.ieee.org/document/9309213; https://doi.org/10.1016/j.isatra.2017.11.010; https://doi.org/10.1016/j.isatra.2021.06.002

  • How have you removed the chattering problem in the simulation results? 
  • The directions to further and improve the work should be added as future recommendation after ‘conclusions’ section.

Considering these comments, I suggest major revision of this paper.

Reviewer 4 Report

The paper is a good one and deserves publication after some improvements.

The authors used the appropriate techniques for analysis of the research objects in order to meet aims of the study. The accurate interpretation of outcomes, well substantiated by the results of the analysis has been achieved by them. The presentation of the results in terms of the research objectives has been successfully made. Appropriate methods have been used in a well-founded manner. 

The results are interesting.

Nevertheless the following points should be clarified before that the paper is published.

Please justify the choice of the Lyapunov functions. All Lyapunov functions which you considered are quite interesting but please try to explaining the inspiring ideas. This could be usefull also for the Reader in other possible applications.

Could Eq. (38) present a singularity or almost a singularity?

Please discuss also this point!

Very good paper which deserves publication!

A multi input sliding mode control for peltier cells using a cold–hot sliding Surface A Mironova et al. Journal of the Franklin Institute 355 (18), 9351-9373

A switching kalman filter for sensorless control of a hybrid hydraulic piezo actuator using mpc for camless internal combustion engines P. Mercorelli  2012 IEEE International Conference on Control Applications, 980-985

Throttle valve control using an inverse local linear model tree based on a fuzzy neural Network M. Nentwig, P. Mercorelli. 2008 7th IEEE International Conference on Cybernetic Intelligent Systems, 1-6

Global finite-time stabilization of planar linear systems with actuator Saturation Y. Su et al. IEEE Transactions on Circuits and Systems II: Express Briefs 64 (8), 947-951

Sun, J. et al.. Fixed-time sliding mode disturbance observer-based nonsmooth backstepping control for hypersonic vehicles. IEEE Trans. Syst., Man, Cybern., Syst. 2020

Zhang, Z.; et al. Multivariable sliding mode backstepping controller design for quadrotor UAV based on disturbance observer. Sci. China Inf. Sci. 2018

Li, B. et al. Extended state observer-based finite-time dynamic surface control for trajectory tracking of a quadrotor unmanned aerial vehicle. Trans. Inst. Meas. Control 2020, 42, 2956–2968.

Round 2

Reviewer 1 Report

I have no further comments

Reviewer 3 Report

I think this paper can be accepted now.